
# The mechanisms and meteorological drivers of the ozone-temperature relationship

William C. Porter[1], Colette L. Heald[2]

[1]Department of Environmental Sciences, University of California, Riverside, CA 92521, USA

[2]Department of Civil and Environmental Engineering, MIT, Cambridge, MA 02139, USA

*Correspondence to*: William C. Porter (william.porter@ucr.edu)

**Abstract.** Surface ozone ($O_3$) pollution levels are strongly correlated with daytime surface temperatures, especially in highly polluted regions. This correlation is nonlinear and occurs through a variety of temperature dependent mechanisms related to $O_3$ precursor emissions, lifetimes, and reaction rates, making the reproduction of temperature sensitivities – and the

projection of associated human health risks – a complex problem. Here we explore the summertime $O_3$-temperature relationship in the United States and Europe using the chemical transport model GEOS-Chem. We remove the temperature dependence of several mechanisms most frequently cited as causes of the $O_3$-temperature "climate penalty", including: PAN decomposition, soil NOx emissions, biogenic VOC emissions, and dry deposition. We quantify the contribution of each mechanism to the overall correlation between $O_3$ and temperature both individually and collectively. Through this analysis

we find that the thermal decomposition of PAN can explain, on average, 20% of the overall $O_3$-temperature correlation in the United States. The effect is weaker in Europe, explaining 9% of the overall $O_3$-temperature relationship. The temperature dependence of biogenic emissions contributes 3% and 9% of the total $O_3$-temperature correlation in the United States and Europe on average, while temperature dependent deposition (6% and 1%) and soil NOx emissions (10% and 7%) also contribute. Even considered collectively these mechanisms explain less than 46% of the modeled $O_3$-temperature correlation

in the United States and 36% in Europe. We use commonality analysis to demonstrate that covariance with other meteorological phenomena such as stagnancy and humidity can explain the bulk of the remainder of the $O_3$-temperature correlation. Thus, we demonstrate that the statistical correlation between $O_3$ and temperature alone may greatly overestimate the direct impacts of temperature on $O_3$, with implications for the interpretation of policy-relevant metrics such as "climate penalty".

**1 Introduction**

Tropospheric ozone ($O_3$)  negatively influences human health, agricultural crop yields, and ecosystem integrity (Monks et al., 2015; World Health Organization, 2006; Tai et al., 2014; Fuhrer et al., 2016). As a secondary pollutant, $O_3$ is not directly emitted from natural or anthropogenic sources, but rather forms as a result of photochemistry in the presence of precursors including nitrogen oxides ($NO_x$), carbon monoxide (CO), and volatile organic compounds (VOCs). While the chemical



processes leading to the formation of tropospheric $O_3$ are well understood, the sensitivity of $O_3$ production to changes in ambient conditions and precursor concentrations are complex and nonlinear. Local $NO_x$ and VOC emissions are two of the most important contributors to daytime tropospheric $O_3$ production, but the ratio between the two can be as important as the overall emission magnitudes themselves (Sillman, 1999). $NO_x$/VOC emission ratios of roughly 1:8 produce the highest $O_3$

production rates in simplified box models (Sillman and He, 2002). Therefore, increases in precursor emissions might increase, maintain, or even reduce $O_3$ concentrations, depending on the initial $NO_x$/VOC ratio.

Further contributing to this complexity, $O_3$ formation and transport are highly sensitive to local meteorological conditions (Elminir, 2005). Precursor emissions and concentrations themselves can depend on the weather, for example in the case of temperature-dependent emission of biogenic VOCs from vegetation (Guenther et al., 1995). As a product of photochemical

reactions, tropospheric $O_3$ formation also requires sunlight, and can be sensitive to atmospheric stability, transport, and mixing conditions. Hot, sunny, stagnant conditions are often associated with the greatest risk of extreme $O_3$ events, as these days typically provide the ideal combination of precursor concentrations, photochemical reactions, and stable conditions for the pollutant to form and persist over an extended period of time (Jacob et al., 1993; Lin et al., 2001).

Because of this sensitivity to climate, increases in continental surface $O_3$ have been identified as a possible negative side

effect of a warming climate, a relationship commonly referred to as the "ozone climate penalty". First coined in 2008 by Wu et al., the climate penalty quantifies the additional ozone present in a warmer environment, as well as the additional anthropogenic emissions reductions necessary to compensate for this enhanced $O_3$ production. Given a 2-5 ppbv increase in $O_3$ expected with 2050 climate projections, Wu et al. concluded that an additional 10% reduction in $NO_x$ emissions would be necessary to mitigate these climate-driven ozone increases, above and beyond the ongoing reduction in $NO_x$ emissions

observed across much of the industrialized northern mid-latitudes. This climate penalty is highly region-specific, depending on both current local conditions as well as the nature of future changes. In related work, Bloomer et al., 2009 defined the slope of the observed daily $O_3$/temperature correlation as the "climate penalty factor", and found a decreasing trend in this factor over time as a result of $NO_x$ emission reduction efforts.

While not synonymous, the long-term climate penalty defined by Wu et al. and the daily climate penalty factor calculated by

Bloomer et al. can be understood to be driven by a similar set of temperature-dependent mechanisms. Previous work has examined this temperature-$O_3$ relationship and identified several mechanisms most likely to be responsible, in particular temperature-dependent biogenic VOC emissions and PAN dissociation rates (Jacob et al., 1993; Sillman and Samson, 1995; Jacob and Winner, 2009). Additionally, the temperature dependence of natural soil NOx emissions (Yienger and Levy, 1995) and $O_3$ dry deposition (Wesely, 1989) have been recognized in previous studies, and could contribute to the overall $O_3$-

temperature correlation. Each of these four mechanisms are included in typical chemical transport models (CTMs) used to study atmospheric chemistry, making these models useful tools for estimating the relative contributions of each mechanism to the overall $O_3$-temperature relationship.



## 2 Model Description

To investigate the relative importance of each temperature-dependent mechanism in governing the overall $O_3$-temperature relationship we explore multiple regional sensitivity cases with the chemical transport model GEOS-Chem v9-02 (www.geos-chem.org ). GEOS-Chem is driven by assimilated meteorology from the NASA Global Modeling and Assimilation Office (GMAO); here we use the GEOS-5 product for 2010-2011. Our simulations over North America and Europe are performed at the native grid horizontal resolution of 0.5°x0.667° with 47 vertical levels. Boundary conditions are provided from a global GEOS-Chem simulation at 2°x2.5° horizontal resolution.

The default tropospheric chemical mechanism in GEOS-Chem v9-02 includes a description of NOx-hydrocarbon-$O_3$-aerosol chemistry with over 120 species which participate in over 400 kinetic and photolytic reactions (Mao et al., 2013). To better capture the temperature dependence of $O_3$ formation as a result of biogenic emissions, we add monoterpene chemistry to the standard GEOS-Chem v9-02 gas-phase mechanism following Fisher et al. (2016), as in Porter et al. (2017). We use the EPA's NEI2005 emissions inventory for anthropogenic emissions over the United States after scaling them up to match NEI2011 national totals for the years 2010 and 2011, then reducing $NO_x$ emissions following the recommendations of Travis et al. (2016). European anthropogenic emissions are taken from EMEP inventories (Auvray and Bey, 2005). To represent global biomass burning we use the GFED3 inventory (Mu et al., 2011). $NO_x$ emissions from lightning are treated using a modified parameterization first developed by Price and Rind (1992) and further constrained by satellite data (Murray et al., 2012). Soil NOx emissions and biogenic hydrocarbon emissions are calculated online following the Hudman et al. (2012) and MEGAN2.1 (Guenther et al., 2012) schemes. Dry deposition is modeled using the Wesely "resistor in series" approach (1989). Wet removal includes contributions from scavenging in convective updrafts, in-cloud rainout and below-cloud washout and is described by Amos et al. [2012].

### 2.1 Ozone-Temperature Mechanisms in the GEOS-Chem Model

The temperature dependence of biogenic VOC emissions (especially those of isoprene) has been frequently cited as an important mechanism contributing to the observed $O_3$-temperature correlation (Wu et al., 2008; Jacob and Winner, 2009; Doherty et al., 2013; Rasmussen et al., 2013), but the magnitude of this biogenic contribution to $O_3$-temperature sensitivity remains uncertain. Additional VOC emissions on hot days would be expected to increase $O_3$ production in areas high in $NO_x$, but other areas – especially those with a particularly low $NO_x$/VOC ratio – might show constant or even reduced $O_3$ levels due to ozone quenching (Loreto and Velikova, 2001), leading to an inverse relationship. Biogenic emissions also do not necessarily vary linearly with temperature. Isoprene emissions, for example, are observed to plateau and eventually shut down completely at very high temperatures (Harley et al., 1999). Representative isoprene and monoterpene emissions response curves are shown in the upper left of Figure 1, based on the GEOS-Chem implementation of MEGAN2.1. In the United States, isoprene and monoterpene emissions are highest in the southeast region, where high temperatures and foliage



density provide ideal conditions in summer months (Figure 2, a and b). Europe is characterized by much lower emissions of isoprene overall, though monoterpene emitters are relatively common across the region (Figure 2, a and b).

While $NO_x$ levels in the lower troposphere are dominated by anthropogenic sources throughout the year, natural processes can also play an important role (Zhang et al., 2003). Of the commonly recognized biogenic sources of $NO_x$, emissions of NO

as a result of microbial activity in the soil has the clearest and most widely observed temperature relationship (Williams et al., 1992). Building upon the work of Hudman et al. (2012) and others, GEOS-Chem includes an exponential temperature-dependent factor for soil $NO_x$ emissions, with plateaus at 30C  (Figure 1, upper right), along with additional factors to account for vegetation type, soil moisture, fertilizer treatment, and canopy losses. This scheme has been shown to produce $NO_2$ levels in broad agreement with satellite observations in terms of spatial and temporal variability, though a systematic

underprediction in model results suggests that modeled soil emissions may need to be further increased overall (Vinken et al., 2014). Modeled summer $NO_x$ emissions vary greatly by location, peaking in the American Midwest and southern European countries, respectively (Figure 2c).

As a so-called $NO_x$ reservoir species, PAN ($CH_3COO_2NO_2$) serves as an important means of nitrogen transport and is one of the primary chemical links between $O_3$ and daytime temperature. A product of reactions between non-methane VOCs and

$NO_x$, PAN has an atmospheric lifetime that is typically longer than its ozone-producing precursors. However, due to the temperature dependence of its primary sink – thermal decomposition – this lifetime varies significantly based on meteorological conditions, with warmer temperatures favoring PAN decomposition and thus local $NO_x$ production (Figure 1, bottom left and Fischer et al., 2014). This temperature-sensitivity has been identified as a dominant reason for the $O_3$-temperature relationship in past measurement and modeling studies (Beine et al., 1997; Dawson et al., 2007; Jacob and

Winner, 2009). PAN concentrations tend to correlate with $NO_x$ emissions, and therefore modeled concentrations peak in the eastern United States as well as central Europe (Figure 2d), where anthropogenic emissions are highest.

Depositional loss to vegetation and other surfaces is a key sink of $O_3$ and other pollutants. Traditional models of dry deposition processes use a "resistor-in-series" approach, in which barriers to $O_3$ deposition through various pathways are parameterized and represented as an electrical circuit (Wesely, 1989). This model has had some success in reproducing

observed patterns $O_3$ deposition velocities, though large remaining uncertainties remain due to the scarcity of long-term measurements (Silva and Heald, 2018). In the Wesely resistance scheme, surface temperature influences deposition rates in two ways: through a stomatal resistance term that is very high at two extremes (typically freezing temperatures and around 40 ℃) and reaches a minimum at some ideal temperature (Figure 1, bottom right), and an exponentially decreasing nonstomatal term designed to reduce deposition over frozen (or nearly frozen) surfaces. In typical summer environments

across the United States and Europe, only the stomatal term is relevant in practice, linking extremely high temperatures with increased stomatal resistance, thereby increasing local $O_3$ levels on very hot days. While observations of $O_3$ dry deposition velocities relative to meteorological drivers show mixed results (Clifton et al., 2017), in principle large-scale increases in stomatal resistance as a result of changes in temperature could lead to increases in $O_3$ concentrations. Summer $O_3$ deposition



velocities across the United States and Europe as simulated by GEOS-Chem tend to range from 0.2 to 0.5 cm/s, depending on local surface type and climatology (Figure 2e).

## 3 Methodology

To represent our control case we use a 2-year base scenario (BASE) for 2010-2011 in which temperature-dependent processes within GEOS-Chem are unchanged. We then sequentially remove the temperature dependence from the four key $O_3$-T mechanisms discussed in Section 2.1 to explore the impact that each has on the overall $O_3$-T relationship over a 3-month summer time period (JJA), with an additional month for spinup. Finally, we run nested regional simulations for each year over the United States and Europe, again discarding the first month of each run to focus on the three summer months (JJA). To isolate the impact of temperature dependence on biogenic emissions (BIO case), dry deposition (DEP case), and soil NOx emissions (SOIL case), we generate a set of hourly temperatures representing the mean summer (JJA) value at each nested grid cell. This averaged diurnal cycle is then substituted into each examined mechanism in turn, resulting in a repeating temperature profile being applied to calculations related to the modified mechanism. Through this procedure, diurnal patterns are preserved while day-to-day temperature variability for that mechanism is removed, preventing it from directly influencing the overall daily $O_3$-temperature correlation. In the PAN case, the default GEOS-Chem chemical mechanism is modified to remove temperature dependence from PAN dissociation by assuming a local constant temperature of 15º C everywhere for that particular reaction.

To confirm that our four chosen mechanisms (biogenic VOC emissions, soil $NO_x$ emissions, PAN dissociation, and dry deposition) are in fact collectively responsible for most of the direct connection between temperature and $O_3$ within GEOS-Chem we perform an additional set of sensitivity tests over each of our regional domains. In one modified case we uniformly increase all temperatures by 1° C, resulting in widespread increases in average surface $O_3$ levels (Figure 3a). In a second modified case we again increase temperature by 1 ºC, but decouple temperature from the four chosen mechanisms using original mean hourly temperatures as described above. In the decoupled case, surface $O_3$ shows negligible differences in mean surface $O_3$ (Figure 3, b and c), indicating that the four decoupled mechanisms dominate the directly modeled $O_3$-T relationship, with the residual $O_3$ changes likely resulting from temperature-dependent chemical kinetics for species other than PAN.

For observational comparison, we use data from the EPA's Air Quality System (AQS) network of monitoring sites (US Environmental Protection Agency), as well as the AirBase air quality database maintained by the European Environment Agency (EEA).



## 4 Results and Discussion

The simulated $O_3$-temperature relationship in GEOS-Chem for the two modeled summers, as represented by the slope of a gridded $O_3$/T OLS regression, is fairly consistent with AQS and AirBase observations, lending confidence to the use of modeled sensitivity comparisons to examine the significance of underlying mechanisms (Figure 4). In both the United States and Europe, spatial patterns and overall mean values of the $O_3$/T correlation are fairly well represented, though the full range of sensitivities is not reproduced in the model output ($R^2$ of 0.40 and 0.42, mean bias of 0.18 ppbv $O_3$ °C$^{-1}$ and 0.06 ppbv $O_3$ °C$^{-1}$ for the United States and Europe respectively). In spite of the relatively strong agreement between modeled and observed $O_3$/T correlations, we highlight a number of shortcomings in the modeled representation of this relationship which may explain the remaining discrepancies between the model and observations. For one, the anthropogenic emission inventories used in GEOS-Chem are independent of daily temperatures, while in reality there are connections between meteorological variability and emissions from human activities such as transportation and energy production. In addition, the grid cell size in GEOS-Chem is incapable of capturing the full diversity of subgrid meteorological phenomena, many of which may be important at the surface station level. Local temperature and $O_3$ fluctuations may vary significantly from those of the gridded average. These issues, among others, may contribute to some of the differences seen in the comparison between observed and modeled sensitivities. In particular, the magnitude of both high and low extremes tends to be underestimated in gridded output from GEOS-Chem, resulting in a tighter distribution of modeled output and skewed slope of modeled vs. observed values, especially in Europe (Fig 4, right).

However, given that the mean values and spatial distribution of regional $O_3$/T sensitivities are generally consistent with observations, we analyze the mechanisms contributing to modeled sensitivities by decoupling them from temperature variability individually and simultaneously. Removing temperature dependence from the four chosen mechanisms has noticeable impacts on correlations between temperature and $O_3$ in the simulated cases, with regional differences apparent in each case. For each of the 4 cases examined, the strength of the $O_3$-temperature dependence (measured via the coefficient of determination $R^2$) was examined through linear regression and compared to that seen in the BASE case. When subtracted from the BASE values, the resulting difference in $R^2$ can be understood as the contribution of that particular mechanism to the overall modeled sensitivity (Figure 5).

Temperature-dependent biogenic VOC emissions have a positive impact on $O_3$-temperature correlation through most of the United States, especially around urban centers, but have a negative impact across much of the southeast. This is consistent with expectations based on $NO_x$/VOC ratios (Figure 2f), in which $NO_x$-rich regions experience a boost in $O_3$ production when rising temperatures lead to additional VOC emissions. Much of the southeast region of the United States, however, is already saturated in VOCs (primarily isoprene), and thus additional emissions on hot days reduce $O_3$ production efficiency, or even act as an $O_3$ sink. The heavily forested northern regions of Europe are likewise less influenced (or even negatively influenced) by the temperature dependence of biogenic emissions, while the high $NO_x$ regions of central and southern



Europe show strong positive contributions. Changes in $R^2$ reach up to 0.14 and 0.21 in the United States and Europe respectively, representing on average 3% and 9% of the overall regional $O_3$/T correlation (Figure 5).

The impact of temperature-dependence in dry deposition is distributed roughly congruent with LAI coverage across the United States, contributing up to 0.14 to the $O_3$/T $R^2$ but only 0.02 on average. Little effect is seen in the heavily forested regions of Northern California and the Pacific Northwest, but since deposition is a removal effect and $O_3$ levels are relatively low in those regions to begin with, changes in deposition rates could be expected to have minimal impact on the overall $O_3$-temperature relationship there. The impact of temperature-dependent dry deposition is even less pronounced in Europe, reaching up to 0.08, but averaging less than 0.01 across the region.

Temperature-dependent soil $NO_x$ emissions contribute around 0.04 to the coefficient of determination in both regions, representing 10% of the total $R^2$ value in the United States and 7% in Europe. Notably, the impact of temperature-dependence in soil emissions does not match up directly with the overall magnitude of those emissions themselves (Figure 2c), indicating that this fluctuation represents a relatively minor and diffuse effect. Areas characterized by lower $NO_x$/VOC ratios due in part to low $NO_x$ emissions (Fig 2f) are also more likely to exhibit stronger sensitivity to temperature-driven soil $NO_x$ variability.

The temperature dependence of PAN decomposition is a strong contributor to the $O_3$-temperature relationship in both the United States and Europe, particularly in the American Midwest, where the positive impact of this mechanism reaches 0.32. Impacts are also visible across most of the eastern United States, as well as California's Southern and Central Valley regions, and the $O_3$/T $R^2$ increases by 0.07 on average in the US (almost 20% of the total mean). PAN temperature sensitivity is also a strong contributor to the $O_3$/T relationship in Europe by up to 0.14 (9% of total mean $R^2$). Of the examined model mechanisms in the United States, PAN lifetime is the strongest overall contributor to the correlation between $O_3$ and temperature, though it places a close second to biogenic emissions in Europe.

While each modeled mechanism contributes to the overall $O_3$/T relationship in the United States and Europe, none of them come close to completely explaining the BASE case correlation between $O_3$ and T. Even when all temperature-dependent mechanisms are removed from the model (the ALL case), most regions still show $O_3$-temperature sensitivities of 50% or more of their original BASE values as measured by $R^2$. While there are uncertainties associated with comparing statistical sensitivities across these simulated cases, it seems clear that the $O_3$-temperature relationship cannot be fully (or even mostly) explained by these 4 mechanisms within GEOS-Chem (Figure 5).

Beyond the directly temperature-dependent emission and loss mechanisms examined within GEOS-Chem, many other meteorological effects can influence surface $O_3$ levels, and correlations between these phenomena and temperature could show up as part of the observed $O_3$-temperature correlation. For example, strong winds can act as a removal mechanism for locally produced $O_3$. If strong winds are also correlated to cooler temperatures, this would show up as a positive correlation between $O_3$ and temperature, despite the lack of any explicit temperature-dependent mechanism. While decoupling other meteorological processes from temperature in the manner demonstrated above can be highly problematic, even within a



model, statistical methodologies such as commonality analysis allow for some degree of attribution of observed predictive power between temperature and the other meteorological drivers (Seibold and McPhee, 1979).

To quantify the contributions of meteorological variables to the modeled $O_3$-temperature correlation, we apply commonality analysis to all gridded output. Through this methodology we are able to decompose all gridded surface $O_3$-temperature $R^2$

values into unique and shared contributions among each of the 5 variables examined: maximum daily temperature (T), humidity (HUM, represented by dew point temperature), mean wind speed (WSPD), wind direction (WDIR), change in mean surface pressure ($\Delta$P), and planetary boundary layer height (PBL). The unique correlations for each of these variables are shown in Figure 6, along with the portion of their correlation shared with any other variables (in the case of T) or shared with daily maximum temperature (in the case of the other 5 meteorological variables).

Each "unique" component represents the portion of explained variability that could be explained solely by one meteorological variable among the 6, meaning that the $R^2$ value would be expected to drop by that amount if the predictor were removed from the linear fitting equation. "Shared" components can be understood as overlap between predictor variable contributions, meaning that the actual mechanism responsible for the correlation might reasonably be attributed to any of the involved predictors. While this methodology is imperfect, especially given the assumption that not all relevant

meteorological processes are represented by these 6 predictors, it does provide additional insight on how and where the $O_3$-temperature correlation might be at least partially explained by correlation with other meteorological phenomena.

As shown, temperature has the strongest and most widespread unique correlation with $O_3$ variability of any of the 6 meteorological variables included in both the United States and Europe. However, even this strong unique contribution is significantly less than the magnitude of the shared component, meaning that collectively the remaining 5 predictors could

potentially explain the majority of the predictive power that temperature offers alone. The overall predictive power for each meteorological variable, along with the respective shared and unique components, can be further visualized through their mean values across all grid cells. Figure 7 shows region-averaged attribution of shared and unique correlation through stacked columns: each individual column height shows the total correlation (in the form of $R^2$) between ozone and a single meteorological variable, while individually shaded sections differentiate unique and shared components. In each

meteorological column that particular variable's unique contribution is at the bottom, and shared contributions are grouped where possible into clusters of two or three total variables for clarity. To best represent the unique contribution of temperature, commonality analysis presented here is performed on the ALL case, with all four chosen temperature dependent mechanisms decoupled. The difference in total correlation between ALL and BASE cases (which are driven by identical meteorology) is then added into the unique temperature contribution, as this gap can be fully attributed to temperature

dependence. Therefore, performing commonality analysis on the BASE case alone would underpredict the unique temperature contribution, since some percentage of variability driven specifically by temperature-dependent mechanisms could also correlate with other meteorological variables. Combining commonality analysis along with the results of the BASE-ALL comparison makes full use of the attribution information contained in each, since any lost correlation with temperature dependent mechanisms turned off can be attributed directly to temperature alone, better constraining the



commonality analysis itself. Through this analysis it is apparent that over half of the $O_3$-temperature relationship in the United States and Europe (shown by the left-most bar in each panel) can be explained through correlation with one or more meteorological covariates, especially wind direction, humidity, and planetary boundary layer height. Europe shows an even stronger overall correlation between temperature and $O_3$, and much of that increase appears to be related to a stronger influence from wind and humidity.

We note that these unique and shared designations are heavily dependent on predictor variable choice and would certainly vary when calculated using a different set of meteorological predictor variables. Uniqueness in these figures should, therefore, be taken as an upper limit estimate, as the inclusion of additional meteorological covariates could demonstrate commonality with temperature where this six-variable set does not. Furthermore, commonality shared between meteorological variables does not imply causation by any one of the members – it only indicates shared statistical predictive power and the possibility of alternative $O_3$-producing mechanisms. However, there are a number of possible mechanisms that could explain some of the predictive power demonstrated by non-temperature variables. Wind speed and direction have perhaps the most straightforward meteorological relationship to $O_3$, and they represent transport of the pollutant either to or away from its source location. Wind speed is generally inversely correlated to high temperatures, and stable conditions are also favorable for the build-up and retention of high $O_3$ concentrations. Depending on local topography and pressure patterns, wind direction can also correlate strongly with changes in temperature, shifting the final destination of polluted air masses from one location to another. Previous work described a relatively small role for these advective mechanisms (Camalier et al., 2007), but the results here suggest that, after temperature-specific mechanisms have been accounted for, wind speed and direction together account for a larger fraction of explained $O_3$ variance than previously suggested. Humidity can influence $O_3$ formation in a number of ways, both directly and indirectly. Water vapor itself participates in competing $O_3$-related effects: water molecules act as $O_3$ sinks by reacting with $O(^1D)$ atoms to produce OH, preventing the excited oxygen from re-generating $O_3$. However, in polluted conditions the OH can then act as an $O_3$ precursor itself, potentially increasing production through reactions with CO and VOCs. These competing effects may explain the relatively weak unique contribution of humidity on average, though the high shared fraction (especially in Europe) suggests that other indirect impacts may be involved, such as correlation with cloud cover or fog. Mixing depth has shown mixed results as a predictor for ozone in past studies as well (Jacob and Winner, 2009), as the impact of PBL variability depends strongly on location and local conditions. Areas with low surface $O_3$ can show positive correlations with mixing layer height due to the entrainment of higher concentrations from aloft, while polluted regimes can show strongly negative correlations due to the higher concentrations of trapped precursors on low PBL days.

Although the specific mechanisms through which the non-temperature meteorological variables are not identified through this statistical methodology, it is apparent that the majority of modeled $O_3$-temperature correlation left unexplained with the decoupling of temperature dependent mechanisms (T/$O_3$ $R^2$ in the ALL case, Figure 5) can itself be explained in principle through covariance with other meteorological variables, indicating that this covariance could explain the residual correlation left over when temperature dependent mechanisms are turned off within GEOS-Chem. While the difference in $O_3$/T



correlation between the BASE and ALL cases show that these temperature dependent mechanisms do indeed strongly influence the $O_3$-temperature correlation across a large portion of the northern United States and southern Europe (Figure 8, top), the remaining correlation makes up the larger overall fraction. Shared explanatory power, as indicated by the Shared contribution of temperature in the ALL case (Figure 8, middle), indicates that covariance with one or more additional

meteorological variables could explain most of the remaining $O_3/T$ correlation (Figure 8, bottom). In this panel, red areas of each column represent the fraction of BASE $O_3/T$ correlation that is lost through the decoupling of temperature-dependent mechanisms, blue areas show the shared fraction of remaining temperature dependence in the ALL case, and the gray region represents remaining $O_3$ variability that is uniquely explained by temperature but unaffected by the 4 described mechanisms. This remaining correlation could be the result of imperfectly chosen meteorological variables, residual temperature

dependence within the model from chemistry or other mechanisms, or other fluctuations in emissions or other inputs that happen to covary with temperature.

While day-to-day $O_3$-temperature variability is a useful and commonly examined metric for estimating future changes in air quality under a warming climate, it presents challenges with respect to the extrapolation of daily variability into long-term trends. For example, areas that exhibit little day-to-day variability in summer temperature over the study period may appear

to be insensitive to climate change, even though the low $O_3$-temperature correlation is simply a result of short-term climatological stability. The temperature perturbation cases described previously and shown in Figure 3a provide some additional information on how the daily sensitivities examined here compare to larger, long term shifts. While the day-to-day correlation between $O_3$ and temperature from all modeled drivers (Figure 9, top, black fill) predicts increases in $O_3$ of around 1.4 ppb with a 1 °C increase in temperature, the increases resulting from the temperature perturbation case (Figure 9, below)

are less than half of this value on average (0.58 ppb in the US and 0.47 ppb in Europe). These perturbation $O_3$ differences are very similar, however, to what might be predicted from temperature dependent mechanisms alone by subtracting the ALL case slopes from those of the BASE case (top panel, grey fill). Together, the consistency of these two outcomes indicates that projections of $O_3$ concentrations under future climate scenarios will be dependent on an accurate representation of temperature dependent meteorology and dynamics, and that models relying on temperature dependent emissions and

chemical mechanisms alone may underpredict the strength of $O_3/T$ sensitivities by over 60%.

Model behavior can be further analyzed through comparison to surface station observations, which reveals a significant difference (P < 0.001) in model skill (as measured by modeled vs. observed daily mean $O_3$) when grouping stations by overall $O_3/T$ correlation as well as by the relative importance of modeled mechanisms (Figure 10). Matching observations from the EPA's AQS network in the United States and the EEA's AirBase dataset for Europe with nearest neighbor grid

cells from GEOS-Chem output shows that model skill tends to be higher in regions characterized with above-average $O_3$-temperature correlation (BASE case $O_3$-temperature $R^2 > 0.42$). While this does not imply that temperature dependent processes are all modeled correctly, it does at least suggest that temperature-based drivers tend to be better captured by the model than other influences on $O_3$ variability. However, splitting observed stations based on the relative importance of internally modeled mechanisms shows that more work may need to be done on these implementations. Grid cells in which

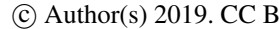

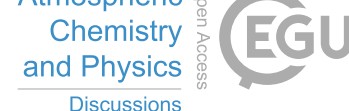

the modeled $O_3$-temperature relationship was dominated by temperature dependent mechanisms (greater than 50% of the $O_3$-temperature correlation lost when temperature dependence was removed in the ALL case) showed much less overall predictive power when compared to the corresponding surface observations (Figure 10, bottom).

## 5 Conclusions

A changing climate implies changes in the physical and chemical regimes governing the emission, formation, and transport of pollutants such as tropospheric $O_3$. Previous work has identified increasing temperatures in particular as a driver of elevated surface $O_3$ concentrations, mitigating the effectiveness of ongoing emissions reductions in the United States and Europe. This means that, under a warming climate, polluted regions would need to cut emissions even further to achieve the same improvement in air quality, adding economic and human health costs to the bottom line of climate change adaptation.

Understanding the mechanisms driving the observed relationship between $O_3$ and temperature is important for guiding improvements in model performance, as well as for better understanding the effects of future changes.

We show here that while temperature-dependent mechanisms such as biogenic emissions and PAN dissociation are often cited as key contributors to the observed $O_3$-temperature relationship, model simulations maintain strong $O_3$-temperature correlations even when these mechanisms are completely decoupled from temperature variability. Analysis of other

meteorological variables suggests that meteorological covariance with temperature may explain a large proportion of $O_3$-temperature correlation – over 40% in the United States and nearly 60% in Europe. The relative importance of covarying atmospheric dynamics indicates that simulations investigating temperature perturbations alone will underestimate overall $O_3$ impacts by a factor of 2 or more, unless temperature-driven changes in other meteorological patterns are also included and accurately represented. Furthermore, comparison with station observations shows that modeled daily $O_3$ values are less

skillful in areas where the $O_3$-temperature correlation is dominated by modeled temperature-dependent mechanisms rather than meteorology, indicating that improved representation of these mechanisms in particular may improve overall model skill with respect to $O_3$ modeling and forecasting.

These results highlight the complexity of pollution projections under changing emissions and climatological conditions, as well as with the attribution of those changes to any individual driver or metric. While surface temperatures can be easily

linked to $O_3$ variability statistically, it is apparent that the robustness of this relationship depends on how consistently coupled those temperature changes continue to be, not only with temperature-dependent physical and chemical drivers of $O_3$ formation, but also with the covarying meteorological patterns that appear to be just as influential. These relationships are further confounded by ongoing changes in anthropogenic emissions, making it especially important to understand the ways in which policy-driven emissions reductions may improve – or fail to improve – air quality within a changing climate.

Ongoing investigations into the importance of these mechanisms, emissions, and atmospheric dynamics will guide future model development, improving forecast skill and better informing policy decision-making.





**Author contribution**

Development of the ideas and concepts behind this work was performed by both authors. Model execution, data analysis, and manuscript preparation were performed by WCP with feedback and advice from CLH.

**Acknowledgements**

This work was supported by the EPA-STAR program (RD-83522801) and a core center grant from the National Institute of Environmental Health Sciences, National Institutes of Health (P30-ES002109). Although the research described in this article has been funded in part by the US EPA through grant/cooperative agreement, it has not been subjected to the Agency's required peer and policy review and therefore does not necessarily reflect the views of the Agency and no official endorsement should be inferred. The authors acknowledge Dr. Brian J. Reich for useful discussions.

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



**Table 1: Summary of GEOS-Chem cases**

| Case | Modifications from default GEOS-Chem |
|------|--------------------------------------|
| BASE | Reduced United States $NO_x$, added monoterpene chemistry |
| BIO | BASE, plus normalized temperature for biogenic VOC emissions |
| SOIL | BASE, plus normalized temperature for soil $NO_x$ emissions |
| DEP | BASE, plus normalized temperature for dry deposition |
| PAN | BASE, plus removed temperature dependence for PAN thermal decomposition |
| ALL | BASE, plus all changes from BIO, SOIL, DEP, and PAN cases |

5  **Table 2: Meteorological variables examined**

| Variable | Description |
|----------|-------------|
| T | Maximum daily temperature |
| HUM | Mean daily vapor pressure (humidity) |
| WDIR | Normalized U and V wind vectors |
| WSPD | Mean daily wind speed |
| $\Delta P$ | Change in daily mean surface pressure |
| PBL | Maximum daily planetary boundary layer height |



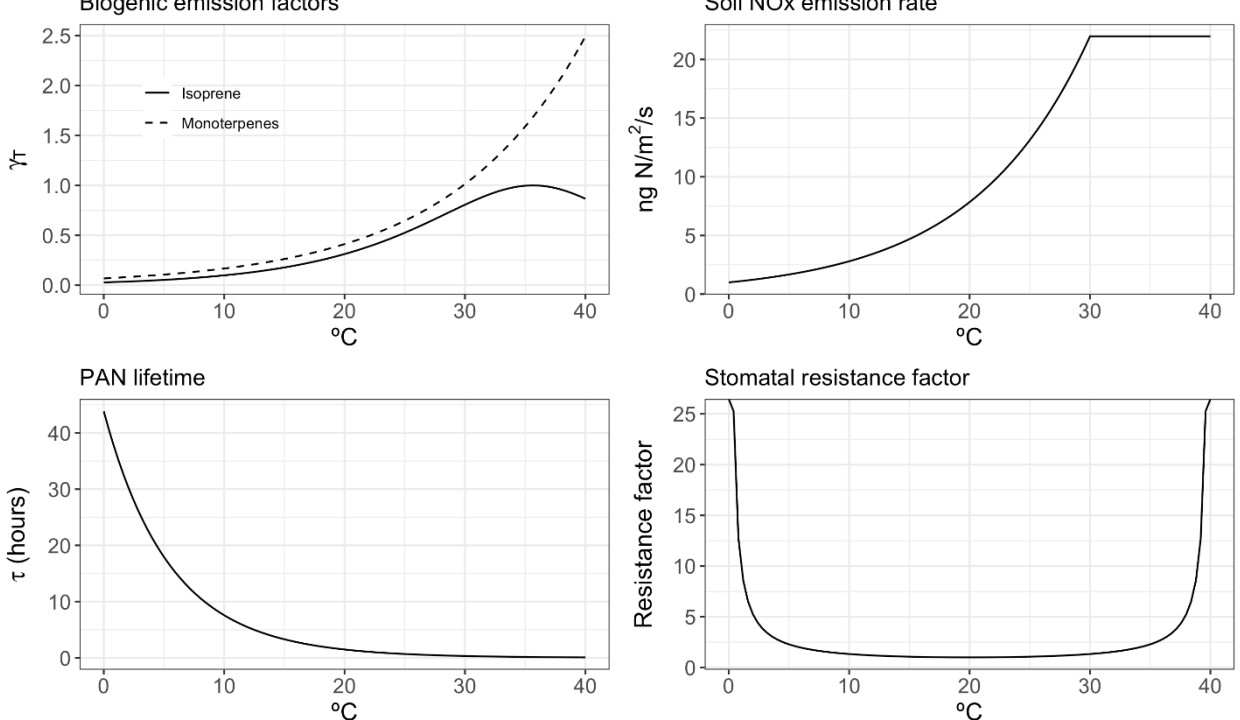

**Figure 1: Representative temperature-dependent mechanism responses within GEOS-Chem for biogenic emissions (top left), soil NOx emissions (top right), PAN lifetime (bottom left), and stomatal resistance (bottom right).**

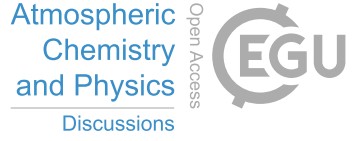



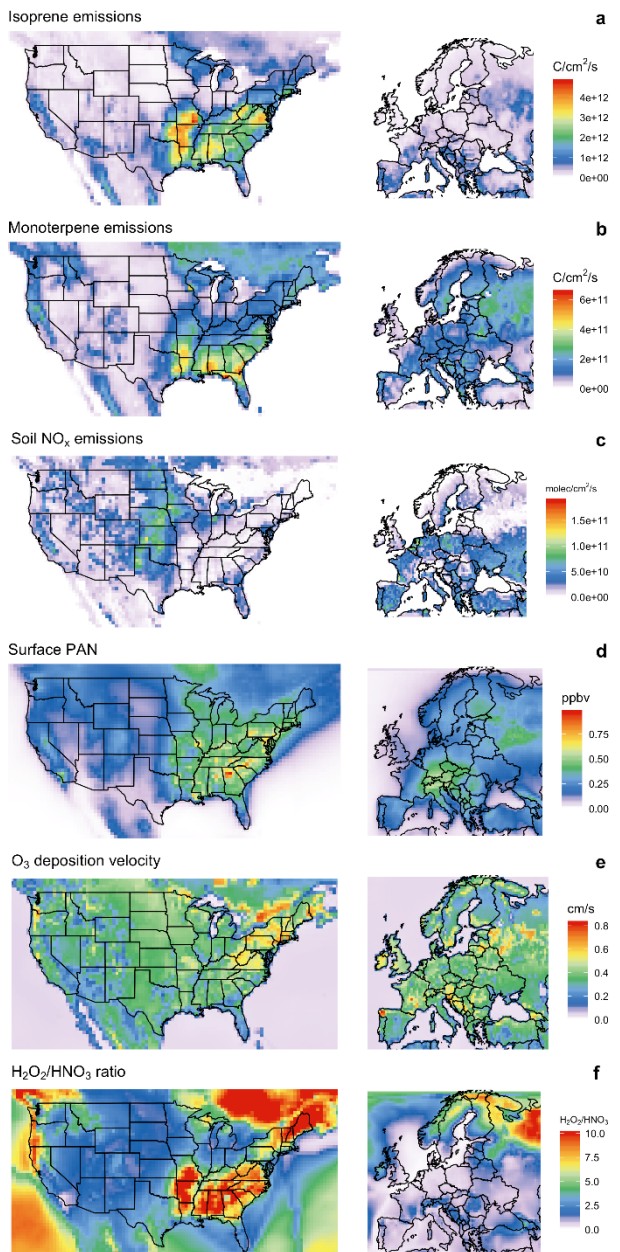

**Figure 2: Summer mean values (JJA 2010-2011) for modeled isoprene and monoterpene emissions (a and b), soil NO$_x$ emissions (c), surface PAN mixing ratios (d), O$_3$ deposition velocity (e), and NOx/VOC sensitivity as represented by the surface H$_2$O$_2$/HNO$_3$ ratio (f).**




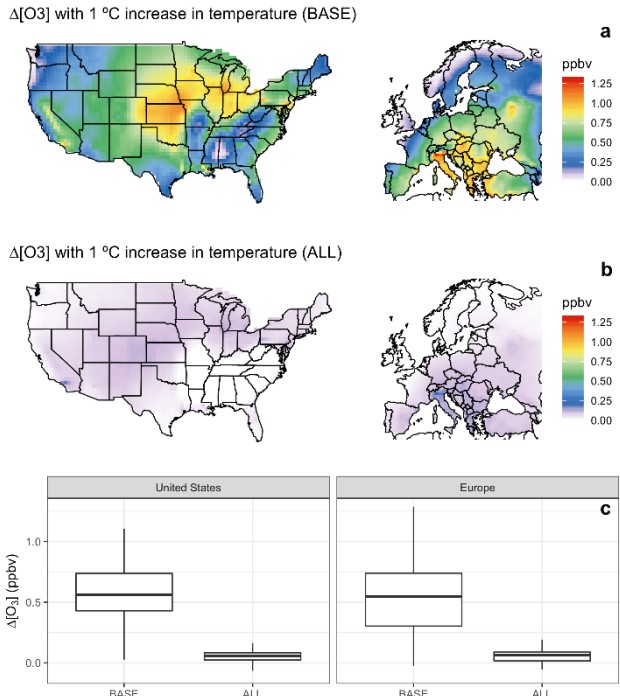

**Figure 3: Increase in O₃ with a 1 ºC increase in temperature in the BASE case (top) and with fixed temperature mechanisms in the ALL case (middle). Distribution of changes for each shown in boxplots (bottom).**





**Figure 4: Regression slopes of summer (JJA) daily maximum 8-hour average O₃ vs. daily maximum temperature for GEOS-Chem and station observations in the United States and Europe.**



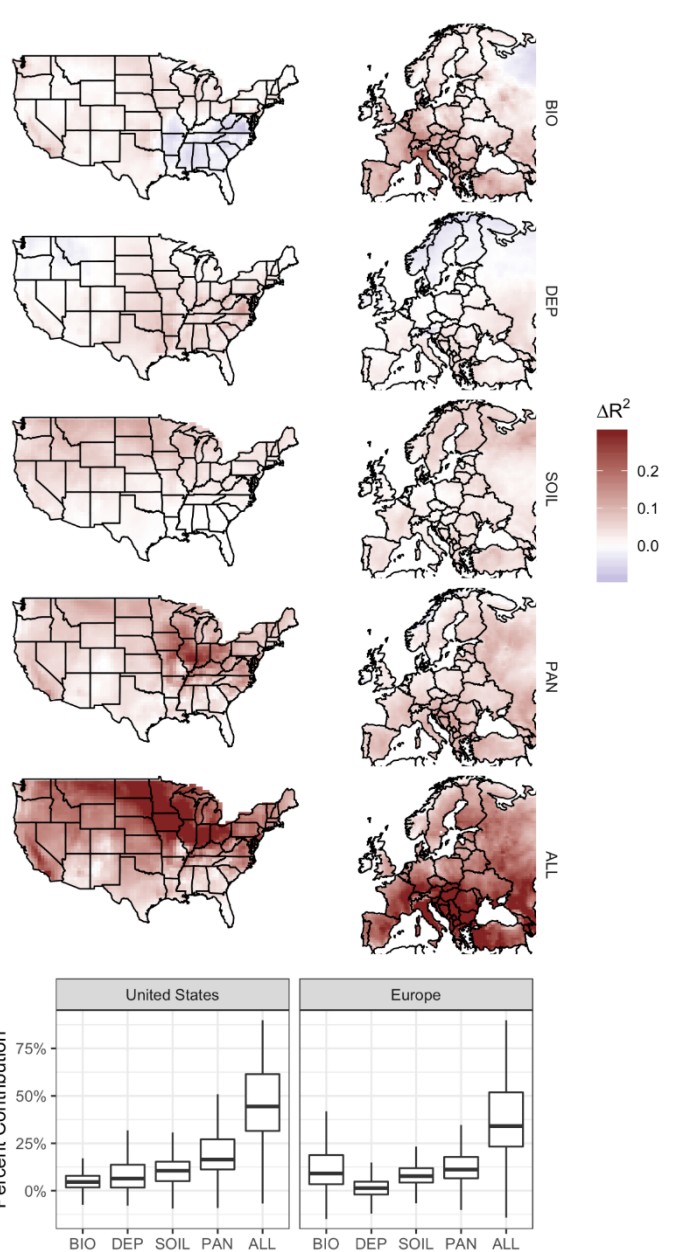

**Figure 5: Impact of temperature dependence of biogenic emissions, O₃ dry deposition, soil NOₓ emissions, PAN lifetime, and all mechanisms at once. Plotted values show the difference between O₃-temperature correlation in the BASE case and that of the modified case in which dependence on daily temperature variability is removed.**





**Figure 6: Unique and shared O₃ correlation among meteorological variables in the BASE case. Unique contributions represent predictive power provided by one meteorological covariate alone, while shared correlation could be attributed to one or more other covariates.**



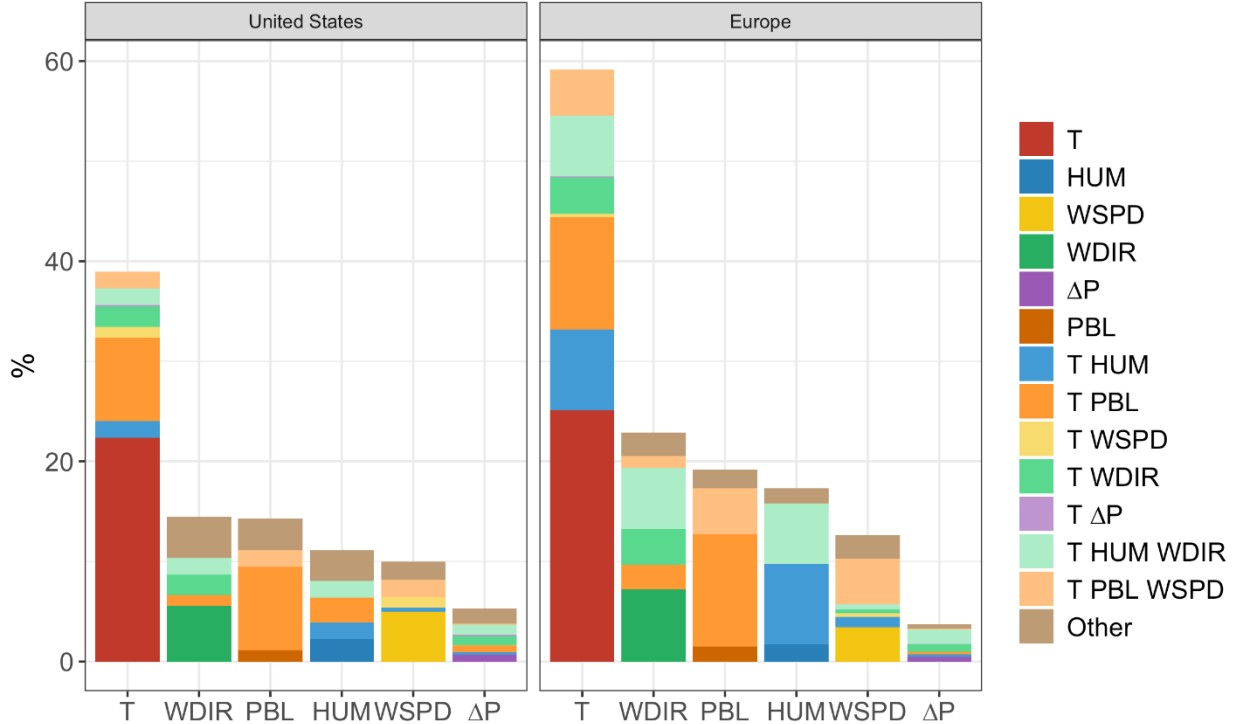

**Figure 7: Unique and shared contributions to O$_3$ correlation for each of 6 different meteorological variables in the BASE case. Column heights represent overall predictive power for each variable, while individual colors indicate predictive power unique to that variable (bottom color in each column) or shared by one or more other meteorological variables.**



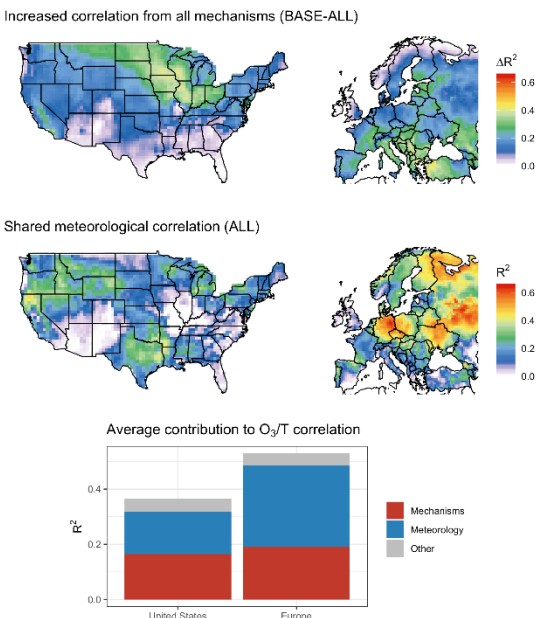

**Figure 8: Total contribution of modeled mechanisms to the O₃-temperature correlation in GEOS-Chem (top), possible contribution of the other included meteorological variables (middle), and mean value for each category by region as a fraction of the total O₃-temperature correlation (bottom).**




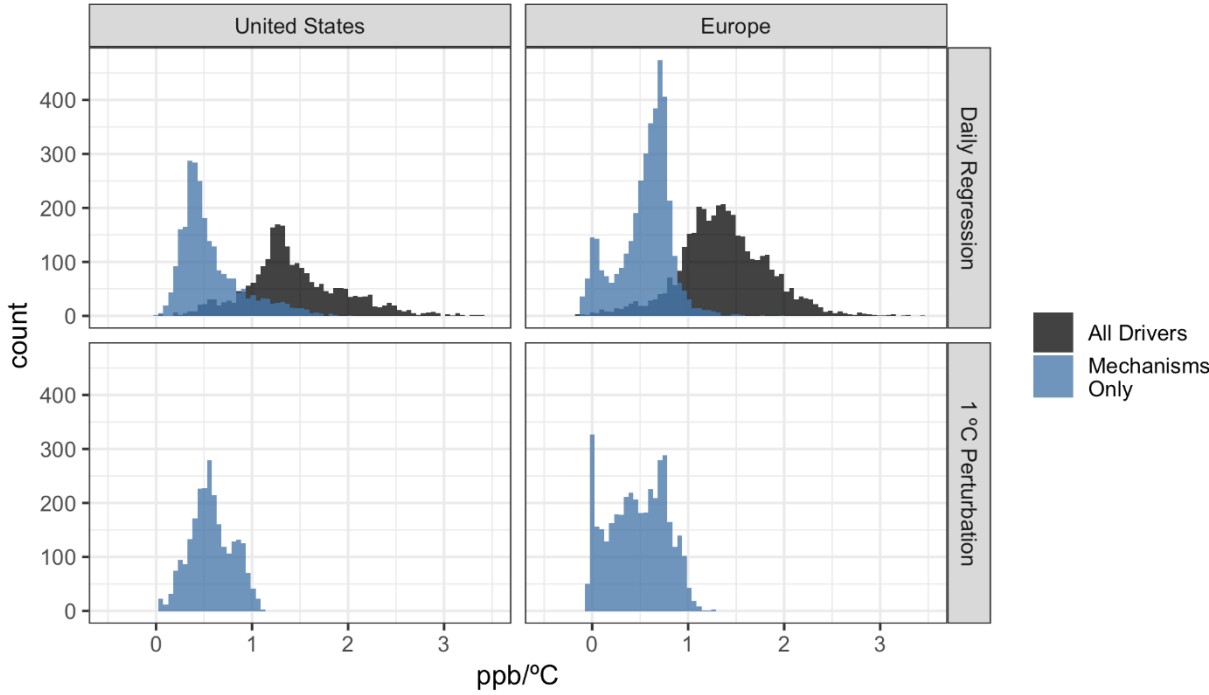

5    **Figure 9: Distribution of O₃/T sensitivities as measured by the slope of OLS regression (above) and mean surface O₃ differences from a flat 1 ºC temperature perturbation (below). Regression values are shown for all modeled drivers (BASE case, black), and the portion of those slopes attributable to temperature-dependent mechanisms (BASE-ALL, grey).**

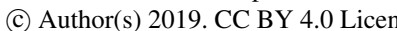



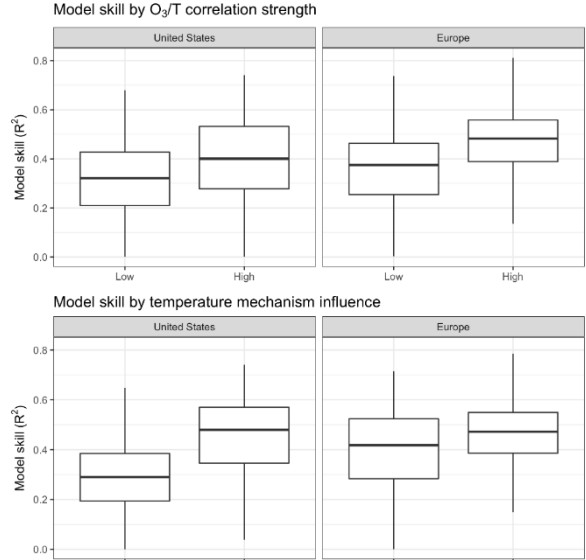

**Figure 10: Differences in model skill compared to surface station observations as a function of overall O₃-temperature correlation (top) and the relative importance of modeled temperature dependent mechanisms (bottom).**