# Peer review of "The mechanisms and meteorological drivers of the summertime ozone-temperature relationship"

_Atmospheric Chemistry and Physics, 2019_

## Referee Comment (RC1) · Anonymous Referee #1 · 5 May 2019

The manuscript provides an interesting investigation on the causes of ozone concentration correlation with atmospheric temperature. This correlation has relevant implications connected to the climate change trends that is expected to increase future surface ozone concentrations. The manuscript provides a useful evaluation of the influence of the temperature depending emissions and chemical model mechanisms in comparison with the other temperature related meteorological forcings. The manuscript is well written and needs just few clarifications. The only part that can be improved concerns the discussion of the model limits that could have an impact on the proposed analysis (see detailed comments).

[Figure]

Detailed comments

2 Model Description Page 3-4 It would be important to add information on limits and uncertainties of the computational schemes implemented in GEOS-Chem for the investigated Ozone-Temperature mechanisms. A general discussion of the known models limits/reliability based on existing literature and of the representativity of the used land and vegetation description datasets would be useful. Page 3 Lines 25-27 The Authors should comment the possible effect of the limited horizontal resolution of their simulations. The dilution of emissions at coarse resolution can cause differences in the simulated O3 concentrations in areas characterized by high NOx emissions, like urban areas, with what would be obtained at higher resolution (e.g. due to titration). Page 4 Lines 1-2 Are the mentioned differences between North America and Europe attributed to the different vegetation species present in the areas? What land-use/landcover datasets have been used? It would be relevant to comment the possible impact of vegetation cover description on the isoprene/terpenes emission (e.g. different broadleaf species can have large differences in isoprene basal emission factors and a "simplified" land cover mapping can cause a relevant bias in BVOC emissions). Lines 8-10 Does this mean that NOx emissions from soil are generally underestimated? Even their impact on the O3-T relationship should be considered underestimated? Lines 31-32 Does this mean that increasing temperature does not necessarily cause an increase of surface layer turbulence but can be associated to reduced turbulence e.g. in stagnation conditions? A reliable vegetation mapping can have relevant impact on dry deposition through z0 and canopy resistance model. 4 Results and Discussion Page 6 Line 3 OLS stays probably for Ordinary Least Square. The acronym has not been previously introduced. Lines 3-17 O3/T slope (Figure 4) seems overestimated for low observed values and underestimated for high observed values. Is there an understandable reason for this model behavior? Page 7 Lines 3-8 The comment suggest that deposition mechanism is scarcely relevant while Figure 5 suggests that its contribution is larger than biogenic emissions and similar to soil emissions in north America, while its contribution is the lowest in Europe. Lines 2-21 Figure 5 lower panel y-axis labels make

difficult for the reader check the % values mentioned in the manuscript text. A different subdivision and maybe figure size would make the manuscript reading easier.

---

## Referee Comment (RC2) · Anonymous Referee #2 · 21 May 2019

** GENERAL COMMENTS** This paper by Porter and Heald investigated an interesting topic, i.e. the processes able to explain the relationship between O3 and surface temperature during summer season in North America and Europe. This relationship is particularly important as a potential metric to define the impact of climate-change to future O3 mixing ratios as well as the required precursor emission mitigation.The topic is well on the focus of ACP and it can be of interest for a wide audience.

Some sections must be clarified before publication. While the discussion of the sensitivity of O3/T regression as a function of the four main processes (BIO, DEP, SOIL, PAN) is rather clear, the application and discussion of the so-called "commonality analysis" is rather confused. I would suggest to describe more in details this methodology in the Section 3, and the providing results in Section 4. Basically, it is not clear what this "commonality analysis" is and how it is performed, actually. References to other study can help the reader. It is not clear how the "unique" and "shared" contributions have been calculated.

A second point that deserve more discussion by the authors is related to the spatial dependence of the agreement between model and observations (figure 4). How these evident disagreements affect the final results and their interpretation/robustness?

**SPECIFIC COMMENTS** METHODOLOGY The model resolution cannot prevent some wrong representation of processes (e.g. emission near urban area or coastal regions). Please comment on this and if available provide references about comparison of model performance with near-surface observations.

Did you perform any basic data check on AQS or AirBase database (e.g. detection of outlier)? Did you try to categorize (especially for the comparison with model) the measurement stations as a function of their altitude, sub-region (e.g. South Europe vs Northern Europe) and type (e.g. urban, rural, remote)? All these attributes can be important when discussing O3-related processes

Pag 5: The description of the methodology from line 9 to 14 is not clear .

RESULTS AND DISCUSSION Pag 6: what " OLS" is?

Figure 4. Significant mismatch exist between model and stations (e.g. over central US).Please better discuss if (and how) you considered this when commenting your results. For Europe, it is almost impossible to appreciate the model results due to the high coverage of stations. Is there any dependence of bias or correlation between model output and observations as a function of station type/elevation?

Figure 9: in the text a "grey" fill is mentioned. In the plots only blues and black are visible. Please improve the colors. Also make the caption more clear: it's difficult to

couple the figure with the text. In general the discussion of Figure 9 is confusing for the reader. Please, start describing what the figure reports (double check with caption) then comment results.

Pag.10: line 20-22: something appears missing in this sentence.
* * *

---

## Author Comment (AC1) · 13 Jul 2019

We thank both referees for their comments and questions regarding our submitted manuscript. Below we have provided a list of concerns raised by each referee, along with our response (in italics) and associated changes or additions to manuscript text (in bold) for each point.

**Referee 1**

1.1 **Limits and uncertainties in GEOS-Chem and input data sets:** 2 Model Description Page 3-4 It would be important to add information on limits and uncertainties of the computational schemes implemented in GEOS-Chem for the investigated Ozone-Temperature mechanisms. A general discussion of the known models limits/reliability based on existing literature and of the representativity of the used land and vegetation description datasets would be useful.

*To provide further background information on the strengths and limitations of tropospheric ozone representation in GEOS-Chem, the following text and citations have been added to the Model Description section:*

**"GEOS-Chem has been shown to reproduce key spatiotemporal features of surface and column ozone observations, though biases and uncertainties are also known (Zhang et al., 2011; Hu et al., 2017). In particular, uncertainties in anthropogenic emission inventories (Travis et al., 2016), various drivers of biogenic emissions (Arneth et al., 2011; Vinken et al., 2014), and lightning NOx (Murray, 2016) have been found to play important roles the variability of tropospheric ozone and its precursors. Uncertainties in spatial inputs, including the datasets used to drive biogenic emissions such as plant functional type and leaf area index distributions, can also influence the resulting biogenic emissions and ozone impacts, and changes or updates to these inputs would influence the magnitude and distribution of the resulting temperature sensitivities (Guenther et al., 2006; Arneth et al., 2011). Ongoing advances in the development of chemical mechanisms relevant to ozone formation and loss (Mao et al., 2013; Sherwen et al., 2016) have also underscored the importance of chemistry. While a full analysis of the sensitivity of the $O_3$/T relationship to each of these factors is beyond the scope of this work, uncertainties in these and other modeled parameters and inputs can all influence both overall ozone production as well as the temperature sensitivities examined here."**

1.2 **Impacts of horizontal resolution coarseness:** Page 3 Lines 25-27 The Authors should comment the possible effect of the limited horizontal resolution of their simulations. The dilution of emissions at coarse resolution can cause differences in the simulated O3 concentrations in areas characterized by high NOx emissions, like urban areas, with what would be obtained at higher resolution (e.g. due to titration).

*Strong spatial gradients in key species such as $NO_x$ can make model resolution an important consideration for many $O_3$-related calculations, such as human health impact estimates (Thompson et al., 2014). However, both temperature and $O_3$ itself tend to be far*

*more regional in distribution and pattern, which we believe makes these fine-scale differences less relevant to our findings here. This is supported by the relatively minor differences apparent in comparisons of the $O_3/T$ relationship in urban and remote locations, both in observed and modeled calculations. (Please see response to 1.7, below.)*

1.3 **Vegetation and other differences between NA and EU:** Page 4 Lines 1-2 Are the mentioned differences between North America and Europe attributed to the different vegetation species present in the areas? What land-use/landcover datasets have been used? It would be relevant to comment the possible impact of vegetation cover description on the isoprene/terpenes emission (e.g. different broadleaf species can have large differences in isoprene basal emission factors and a "simplified" land cover mapping can cause a relevant bias in BVOC emissions).

*Uncertainties in landcover datasets used to drive biogenic emissions and deposition can indeed play significant roles in modeled $O_3$ variability* (Arneth et al., 2011)*. These differences in relative BVOC abundances in the US and Europe are indeed mostly driven by differences in the input plant functional type and leaf area index datasets. This land-use data driving biogenic emissions in GEOS-Chem are derived from those of the MEGAN model, which themselves come from a variety of surface observation and remote sensing products, as described in Guenther et al., 2006 (see table 4 of this publication). A note mentioning the significance of these datasets and their uncertainties has been added. (See response to 1.1 above.)*

1.4 **Underestimation of soil NOx:** Page 4 Lines 8-10 Does this mean that NOx emissions from soil are generally underestimated? Even their impact on the O3-T relationship should be considered underestimated?

*Vinken et al., 2014 found calculated a top-down emissions inventory that was 4-35% higher in magnitude in total than that of the GEOS-Chem a priori inputs, with regional variability in both the magnitude and sign of this difference. However, the impact of this possible underprediction of global soil $NO_x$ on the $O_3/T$ relationship is unclear, and would likely depend on the details of the spatial distribution and seasonal/diurnal timing of the difference. More work is necessary on this topic to further explore the impacts of soil $NO_x$ inventory changes such as these.*

1.5 **Heat and stagnation:** Page 4 Lines 31-32 Does this mean that increasing temperature does not necessarily cause an increase of surface layer turbulence but can be associated to reduced turbulence e.g. in stagnation conditions?

*While increased surface temperatures are associated with more vertical mixing, increased mixing depth has a complex relationship with surface ozone levels. Free troposphere ozone levels are often significantly higher than background surface $O_3$ levels, making vertical*

*mixing less effective at reducing O₃* (Jacob and Winner, 2009). *For this reason, horizontal transport, as measured by wind speed, is considered more important for defining stagnant conditions.*

1.6 **Introduce OLS:** 4 Results and Discussion Page 6 Line 3 OLS stays probably for Ordinary Least Square. The acronym has not been previously introduced.

*The full form of this acronym has been added to the text in question.*

1.7 **Model vs. obs differences in O₃/T:** Lines 3-17 O3/T slope (Figure 4) seems overestimated for low observed values and underestimated for high observed values. Is there an understandable reason for this model behavior?

*It does appear that the extremes of the O₃/T sensitivity distribution are not fully captured within the model. While we cannot conclusively explain this discrepancy here, we discuss several possibilities in the text, which we highlight here:*

"In spite of the relatively strong agreement between modeled and observed $O_3/T$ correlations, we highlight a number of shortcomings in the modeled representation of this relationship which may explain the remaining discrepancies between the model and observations. For one, the anthropogenic emission inventories used in GEOS-Chem are independent of daily temperatures, while in reality there are connections between meteorological variability and emissions from human activities such as transportation and energy production. In addition, the grid cell size in GEOS-Chem is incapable of capturing the full diversity of subgrid meteorological phenomena, many of which may be important at the surface station level. Local temperature and $O_3$ fluctuations may vary significantly from those of the gridded average. These issues, among others, may contribute to some of the differences seen in the comparison between observed and modeled sensitivities. In particular, the magnitude of both high and low extremes tends to be underestimated in gridded output from GEOS-Chem, resulting in a tighter distribution of modeled output and skewed slope of modeled vs. observed values, especially in Europe (Figure 4b)."

*We do appreciate the concern over the failure of the model to capture observed extremes, but would like to underscore that the overall magnitudes of O₃/T correlation between observations and modeled output are very similar, as shown in the new panel in Figure 4. Thus, while the low and high tails of station sensitivities are not fully captured, for reasons that may include those we list above, we believe that the relatively good agreement in mean sensitivities is indicative of the model's ability to capture the key features of the O₃/T relationship fairly well, at most locations. See also comment 2.2 below.*

1.8 **Relevance of deposition mechanism:** Page 7 Lines 3-8 The comment suggest that deposition mechanism is scarcely relevant while Figure 5 suggests that its contribution is

larger than biogenic emissions and similar to soil emissions in north America, while its contribution is the lowest in Europe.

*On an overall percent contribution basis, deposition appears to be the least significant in the Europe, and the second least significant in the US. That said, the text has been modified to make it clear that deposition is a strong contributor to the overall $O_3/T$ relationship in some locations:*

"The impact of temperature-dependence in dry deposition is distributed roughly congruent with LAI coverage across the United States, contributing up to 0.14 to the $O_3/T$ $R^2$ but only 0.02 on average. Little effect is seen in the heavily forested regions of Northern California and the Pacific Northwest, but since deposition is a removal effect and $O_3$ levels are relatively low in those regions to begin with, changes in deposition rates could be expected to have minimal impact on the overall $O_3$-temperature relationship there. **Relative contributions of deposition on a local basis, however, can represent over one quarter of the overall $O_3/T$ correlation in some US locations.**"

1.9    **Figure 5 label sizes:** Lines 2-21 Figure 5 lower panel y-axis labels make difficult for the reader check the % values mentioned in the manuscript text. A different subdivision and maybe figure size would make the manuscript reading easier.

*Label sizes and y-axis tick spacing have both been adjusted to improve readability.*

**Referee 2**

2.1 **Commonality analysis and output processing:** While the discussion of the sensitivity of O3/T regression as a function of the four main processes (BIO, DEP, SOIL, PAN) is rather clear, the application and discussion of the so-called "commonality analysis" is rather confused. I would suggest to describe more in details this methodology in the Section 3, and the providing results in Section 4. Basically, it is not clear what this "commonality analysis" is and how it is performed, actually. References to other study can help the reader. It is not clear how the "unique" and "shared" contributions have been calculated.

*Additional description of commonality analysis has been added:*

"While decoupling other meteorological processes from temperature in the manner demonstrated above can be highly problematic, even within a model, statistical methodologies such as commonality analysis allow for some degree of attribution of observed predictive power between temperature and the other meteorological drivers (Seibold and McPhee, 1979). **Derived from the analysis of linear regression output, commonality analysis involves the calculation of $R^2$ values for all possible permutations of predictor variables included in the analysis. These $R^2$ values are then compared, allowing for the calculation of explained variability that is uniquely provided by one variable or another, along with explained variability that is shared between two or more of the covariates. For the purposes of this study, "unique" refers to that portion of a variables correlation with the response variable (ozone) that is not shared with any other predictor, while "shared" refers to the portion of the correlation that could be attributed to multiple predictors. A more detailed explanation of the equations involved, as well as examples of their application, can be found in Seibold and McPhee, 1979.**

2.2 **Model vs. observations for $O_3$/T comparison:** A second point that deserve more discussion by the authors is related to the spatial dependence of the agreement between model and observations (figure 4) . How these evident disagreements affect the final results and their interpretation/robustness?

*The differences between observed and modeled $O_3$/T sensitivity in some locations certainly suggests that some processes are not being accurately represented, whether due to incomplete or erroneous representation, or subgrid effects that are not adequately described at the modeled resolution. We suggest some explanations for these discrepancies (see 1.7, above) and further use the results of our analyses to try and pinpoint the type of temperature dependence most associated with model/observation differences. Further discussion of the possible implications of these discrepancies has been added:*

In particular, the magnitude of both high and low extremes tends to be underestimated in gridded output from GEOS-Chem, resulting in a tighter distribution of modeled output and

skewed slope of modeled vs. observed values, especially in Europe (Figure 4c). **However, in spite of the notable differences between modeled and observed $O_3$/T relationships at the tails of the distributions, a relatively small overall bias is apparent across station types in both urban and remote regions (Figure 4b). Here, the more remote stations associated with the National Park Service (NPS) are separated from the rest of the AQS dataset for comparison over the United States, while AirBase stations in Europe are split by station area category (Urban/Suburban and Rural). In each category, nearest-neighbor grid cells effectively capture the center of the observed distribution, even though extremes are not fully represented.**

2.3 **Model resolution and performance:** METHODOLOGY The model resolution cannot prevent some wrong representation of processes (e.g. emission near urban area or coastal regions). Please comment on this and if available provide references about comparison of model performance with near-surface observations.

*Please see the response to comment 1.2, above.*

2.4 **Data check and station categorization:** Did you perform any basic data check on AQS or AirBase database (e.g. detection of outlier)? Did you try to categorize (especially for the comparison with model) the measurement stations as a function of their altitude, sub-region (e.g. South Europe vs Northern Europe) and type (e.g. urban, rural, remote)? All these attributes can be important when discussing O3-related processes

*While individual daily values were not filtered beyond the quality control provided by the databases themselves, stations were filtered for dataset completeness (at least 90 days of data required). In response to the reviewer's comment, additional categorization was added to Figure 4 to distinguish between station type using AirBase categories (Urban/Suburban vs. Rural) and AQS source data (separating NPS stations from more urban sites). These differentiations are described in text (see response 2.2), and we see little difference in the model-measurement comparisons between them. The significance of site elevation was explored by dividing stations into elevation terciles representing the lowest, middle, and highest elevation sites (see figure below). While higher elevation stations showed slightly higher responses to temperature, the effect at most stations appears small and the comparison between observed and modeled $O_3$/T sensitivities does not appear to be sensitive to this distinction.*

[Figure]

2.5 **Average diurnal cycle generation:** Pag 5: The description of the methodology from line 9 to 14 is not clear .

*More detail on the generation of mean diurnal cycles has been added:*

To isolate the impact of temperature dependence on biogenic emissions (BIO case), dry deposition (DEP case), and soil NOx emissions (SOIL case), we generate a set of hourly temperatures representing the mean summer (JJA) value at each nested grid cell. **To do so, we generate mean hourly temperatures for each modeled grid cell by averaging each hour (0 through 23) across the 3 modeled months.**

2.6 **OLS introduction:** RESULTS AND DISCUSSION Pag 6: what " OLS" is?

*This acronym (ordinary least squares) has been clarified and defined.*

2.7 **Spatial dependence of model skill for O₃/T comparison:** Figure 4. Significant mismatch exist between model and stations (e.g. over central US).Please better discuss if (and how) you considered this when commenting your results. For Europe, it is almost impossible to appreciate the model results due to the high coverage of stations. Is there any dependence of bias or correlation between model output and observations as a function of station type/elevation?

*The station points have been reduced in size to help alleviate the overplotting issue in Figure 4. Furthermore, a density plot comparison has been added to this figure to underscore the key takeaway from this comparison: while extreme values at the low and high end of the sensitivity distribution are not fully captured, overall mean O₃/T correlation magnitudes are virtually identical between model and observations. Station type has been added in the form of colored points in the scatter plot portion of Figure 4, though there is no appreciable difference in the behavior of the available categories. Elevation has been explored (see response to 2.4 above) but has been ruled out as a*

*dominant explanation for differences between modeled and observed $O_3$/T correlations.*

2.8   **Figure 9 appearance and description:** Figure 9: in the text a "grey" fill is mentioned. In the plots only blues and black are visible. Please improve the colors. Also make the caption more clear: it's difficult to couple the figure with the text. In general the discussion of Figure 9 is confusing for the reader. Please, start describing what the figure reports (double check with caption) then comment results.

*Figure 9 has been revised and the description rewritten:*

**"Figure 9 shows the distribution of $O_3$/T sensitivities (top panels), both as a whole (black fill) and considering only the 4 key mechanisms previously examined (blue fill). These distributions can then be compared to the distribution of $O_3$ changes apparent with a simple temperature perturbation (bottom panels), which intrinsically includes no other meteorological covariance. While the day-to-day correlation between $O_3$ and temperature from all modeled drivers (Figure 9, top, black fill) predicts increases in $O_3$ of around 1.4 ppb for a 1 ºC increase in temperature, roughly half of that is attributable to the examined mechanisms alone. This portion attributed to mechanisms alone is consistent with the mean change in $O_3$ observed from a 1 ºC increase in temperature (0.58 ppb in the US and 0.47 ppb in Europe)."**

2.9   **Sentence clarity:** Pag.10: line 20-22: something appears missing in this sentence.

*This sentence has been revised completely – see 2.8 above.*